# Wild Blueberry Extract Intervention in Healthy Older Adults: A Multi-Study, Randomised, Controlled Investigation of Acute Cognitive and Cardiovascular Effects

**DOI:** 10.3390/nu16081180

**Published:** 2024-04-16

**Authors:** Nancy Cheng, Katie L. Barfoot, Romain Le Cozannet, Pascale Fança-Berthon, Daniel J. Lamport, Claire M. Williams

**Affiliations:** 1School of Psychology and Clinical Language Sciences, University of Reading, Reading RG6 6ES, UK; nancy.cheng@pgr.reading.ac.uk (N.C.); katie.barfoot@reading.ac.uk (K.L.B.); daniel.lamport@reading.ac.uk (D.J.L.); 2Givaudan France Naturals, 250 rue Pierre Bayle, BP 81218, 84911 Avignon, France; romain.le_cozannet@givaudan.com (R.L.C.); pascale.fanca-berthon@givaudan.com (P.F.-B.)

**Keywords:** blueberries, blueberry extract, cognition, cognitive function, flavonoids, anthocyanins, blood, blood pressure

## Abstract

Background: Circadian and homeostatic declines in cognitive performance are observed during the day, most commonly at 14:00. Additionally, postprandial reductions in cognitive ability have been widely demonstrated 1 h after lunch consumption, affecting domains of executive functioning (EF), episodic memory (EM), and attention. Existing evidence shows that anthocyanin-rich foods such as berries may improve or attenuate the decline in EF and EM in ageing adults. Further research is required to assess whether extracts such as wild blueberry extract (WBE) may be beneficial for cognitive function across an acute timeframe, including known periods of reduced functioning. Objectives: (1) Study 1: ROAB: To investigate the efficacy of WBE in maintaining EF and EM throughout the day alongside measures of cardiovascular outcomes in healthy older adults. A range of WBE doses were utilised to identify the optimal dose at which cognitive and cardiovascular effects occur. (2) Study 2: BEAT: To replicate alleviation of cognitive decline during a predicted post-lunch dip whilst also improving cardiovascular outcomes following acute WBE 222 mg supplementation. Methods: Both studies employed a randomised, double-blind, cross-over, placebo-controlled design to explore the effects of WBE intervention versus placebo on several outcomes, including EM, EF, blood pressure, and heart rate in a healthy older adult population (aged 68–75). In ROAB, 28 participants received a single dose of WBE 111 mg, 222 mg, 444 mg, or 888 mg or placebo over a 5-week period, each separated by a 1-week washout. Outcomes were measured at 0 h, 2 h, 4 h, and 6 h post intervention, with intervention occurring immediately after baseline (0 h). In BEAT, 45 participants received WBE 222 mg and placebo (1-week washout). Outcomes were measured at 0 h and 6 h (14:00) when a post-lunch dip was anticipated. This was further enhanced by consumption of lunch 1 h prior to cognitive testing. The WBE 222 mg intervention aligned with known peaks in plasma blueberry polyphenol metabolites at 2 h post dosing, which would coincide with a predicted drop in post-lunch performance. Results: ROAB: A significant dip in executive function was apparent at the 4 h timepoint for placebo only, indicating attenuation for WBE doses. Strikingly, WBE 222 mg produced acute reductions in both systolic and diastolic blood pressure compared with placebo. BEAT: EF reaction time was found to be significantly faster for WBE 222 compared to placebo at the predicted post-lunch dip (14:00), with no other notable benefits on a range of cognitive and cardiovascular outcomes. Conclusion: These two studies indicate that WBE may have cardiovascular benefits and attenuate the natural cognitive decline observed over the course of the day, particularly when a decline is associated with a circadian rhythm-driven postprandial dip. However, it is important to acknowledge that effects were subtle, and benefits were only observed on a small number of outcomes. Further research is required to explore the utility of WBE in populations already experiencing mild cognitive impairments.

## 1. Introduction

The cognitive and mood effects of flavonoid-rich foods have been investigated across the human lifespan in healthy and cognitively impaired patient populations. Improvements in vascular function have been hypothesized to underlie the improvement in cognitive performance, for example, through changes in cerebral blood flow (CBF) [1,2,3,4]. Blueberries have received significant attention in the literature, with benefits evident across different populations, intervention types, durations, and cognitive tasks (for reviews, see [5,6,7,8,9]). Recently, there has been interest in extracts and whether these may provide similar benefits to their whole fruit counterparts. A chronic blueberry, parallel-groups intervention study examined the cognitive effects of two doses of whole fruit wild blueberry powder (WBP 500 containing 1.35 mg anthocyanins; WBP 1000 containing 2.7 mg anthocyanins) and a wild blueberry extract (WBE 111 containing 7 mg anthocyanins) in healthy older adults with self-reported memory complaints [10]. Significant improvements in delayed word recognition memory followed a 3 month daily dose of WBE 111 treatment alongside significant reductions in systolic blood pressure (SBP) at this dose and timepoint, which were found to be sustained at 6 months. Interestingly, neither dose of wild blueberry powder affected cognitive performance or cardiovascular measures following 3- or 6-month treatment. The authors conclude that this may have been due to a lack of EM sensitivity to the small anthocyanin dosages within the powders. Although the WBE dose was small, the extract formulation contained additional antioxidants (L-cysteine and L-glutathione) to enhance the stability of blueberry anthocyanins and increase bioavailability. Indeed, glutathione has been shown to effectively stabilise blackberry anthocyanins [11] and anthocyanins from extracts of grape, black carrot, red cabbage, blackberry, blackcurrant, and cranberry [12]. Further work is therefore required to identify and show whether additional stabilising agents may be an effective way to induce high bioactivity and subsequent benefits using a low product dose. It is possible that the addition of stabilising agents to anthocyanin-rich extracts may cause more effective results over an acute postprandial timeframe rather than over several weeks.

Research exploring acute effects has shown time- and dose-dependent anthocyanin metabolite peaks and associated improvements to vascular function following blueberry treatment at 1–2 h and 6 h post consumption [13]. A single published trial in healthy older adults demonstrated the acute cognitive benefits of a one-off blueberry intervention (30 g freeze-dried blueberry powder) at 2 h post consumption, with no cognitive change detected at 5 h post consumption [14]. Considering known metabolite peaks, the 5 h time point may have been too early to detect cognitive change. Other work from our laboratory has shown that the cognitive effects of blueberry treatment in children [15,16,17,18] and middle-aged adults [19] coincide with polyphenol metabolite peaks in plasma between 1–2 h and 6 h post consumption, highlighting these as potentially sensitive windows for physiological and cognitive change. It would also be reasonable to expect the amplitude of these peaks to be increased by stabilisation of anthocyanins in WBE.

Moreover, examination of key timepoints at which cognitive performance worsens across the course of a day may reveal potentially sensitive periods where flavonoids may exert the most cognitive benefit. Such dips in cognition have been identified as due to circadian and homeostatic rhythm changes [20,21]. Studies revealing a daily dip in alertness and attention have most commonly recorded occurrence at 14:00 h [22,23]. Additionally, a postprandial decline in cognitive performance has been associated with and accentuated by the consumption of lunch [24,25,26]. The post-lunch period, in conjunction with 14:00 h, is therefore a timepoint of interest to examine whether the natural decline in cognitive performance can be attenuated by nutritional intervention, specifically WBE.

Interestingly, in older adults, circadian and homeostatic effects appear to modulate cognitive performance according to task difficulty. Circadian fluctuations in sustained attention are generally less pronounced in older adults compared with young adults, and the amplitude of circadian changes in attention is much lower for the more difficult aspects of a task [27]. These findings are interesting when combined with previous results that have identified flavonoid sensitivity under conditions of high cognitive demand [16,17,19,28,29]. Additionally, berry treatments have shown to be effective at alleviating the natural temporal decline in cognition across the course of a day in children [16], young adults [28], and middle-aged adults [29]. These findings collectively support the growing perception that flavonoid intervention may invoke the highest cognitive benefit in populations or environments where there is cognitive depletion. Examining blueberry treatment under maximally depleting conditions may therefore reveal optimal sensitivity. Hence, a cognitively ageing population at temporally fatiguing points of the day and using a demanding cognitive battery was utilised to this effect in both studies within this paper.

Considering this, the overarching aim of both studies in the current paper was to capture cognitive, mood, and cardiovascular data across two sensitive windows (1–2 h and 6 h) following a range of flavonoid dosages to understand the dose and temporal profile at which WBE flavonoids combined with stabilising antioxidants (L-cysteine and L-glutathione) may elicit cognitive improvement throughout the day. This timeframe coincides with the trajectory of blueberry anthocyanin peak absorption and metabolism [13]. Two healthy older adult samples were recruited independently, representing individuals who are within a sensitive window of cognitive decline and thus an effective point at which preventative interventions may be particularly potent.

The primary objective of ROAB (Study 1) was to investigate the efficacy of WBE in maintaining episodic memory throughout the day in older adults (68–75 years). Dose–response relationships between WBE and cognitive and cardiovascular outcomes have not been previously examined; thus, a range of WBE doses (111 mg, 222 mg, 444 mg, and 888 mg with L-cysteine and L-glutathione) were investigated in order to identify the optimal dose at which cognitive and cardiovascular effects occur and to establish whether these effects are dose-dependent. The lowest dose used here (WBE 111 containing 7 mg anthocyanins) was the dose previously established to produce beneficial cognitive effects and reduction in SBP following 3 months daily treatment [10]. As suggested by a recent meta-analysis on dietary flavonoids and cognition [30], it was predicted that cognitive effects would increase dose-dependently and be most pronounced for the highest dose of WBE (WBE 888 containing 56 mg anthocyanins). The secondary objective of ROAB was to investigate the effect of WBE in maintaining executive function and mood. Acute effects of blueberry supplementation on mood in older adults have not been observed [10,31], so we sought to identify whether there is WBE-related mood change across the day. The tertiary objective of ROAB was to determine the cardiovascular effects following WBE in healthy older adults through measurement of systolic and diastolic blood pressure and heart rate. It was predicted that both measurements would be reduced following WBE compared to placebo, specifically at 2 h and 6 h post intervention, where vasodilatory effects have shown to peak previously [13]. The BEAT trial (Study 2) followed on from the findings of ROAB to further explore a specific dosage (WBE 222 mg, chosen for the significant effects on blood pressure shown in ROAB) and target a timepoint of interest (14:00 h) indicative of when the post-lunch dip in cognitive performance occurs, using cognitive tasks that showed sensitivity to WBE treatment in ROAB. The secondary objective of BEAT was to replicate reductions in SBP and DBP following WBE 222 shown in ROAB.

## 2. Materials and Methods

### 2.1. Inclusion and Exclusion Criteria

Participants across both ROAB and BEAT trials were healthy older adults recruited via the University of Reading Psychology Department’s Older Adult Panel, local community noticeboards, University of the Third Age, and other older adult activity groups in Reading, U.K., and the surrounding area. Participants were eligible to enrol if they were between 68–75 years old and willing to maintain their habitual diet and exercise habits over the course of the study. Participants were ineligible to enrol if they met any of the following exclusion criteria: (i) history of metabolic disorder, diabetes, substance abuse, or diagnosis of psychiatric or neurological conditions; (ii) smokers; (iii) vegetarians; (iv) BMI ≥ 30; (v) allergy to compounds in the investigational product; (vi) > 2 alcoholic beverages per day on average in a week; (vii) use of complementary or alternative medicine for memory or cognitive performance within 1 month prior to study participation; (viii) participation in other clinical trials within the previous month or other cognitive trial within the previous 6 months.

### 2.2. ROAB—Study 1

#### 2.2.1. Participants

An a priori power analysis based on RAVLT delayed word recognition results from Whyte et al. [10] using 111 mg WBE (Cohen’s d = 0.3) indicated that 27 participants would be required to detect a significant effect following WBE compared with placebo. As in Figure 1, 40 volunteers attended a familiarisation visit, after which 12 withdrew due to the demands of the task battery and length of the test day. Twenty-eight participants progressed to randomisation and completed all test stages.

#### 2.2.2. Design

The study employed a randomized, double-blinded, placebo-controlled, cross-over design comprised of 7 visits in total, including 2 practice visits (1 and 2 weeks prior to test day 1) for screening and familiarisation purposes. Questionnaires to assess eligibility were completed at practice visits as detailed in the Measures Section. Following practice visits, participants were randomised to a treatment order by Givaudan France Naturals (250 Rue Pierre Bayle, BP 81218, 84911 Avignon, France) using a computerised random number generator [32]. Investigators and participants remained blind to the relationship between participant number and order of doses. Participants attended the lab on 5 separate occasions to complete each intervention arm, separated by a 7-day washout period. On each intervention day, participants consumed a standardised breakfast and lunch and attended 4 test sessions across the day at set times (see Figure 2).

#### 2.2.3. Intervention

Four doses of a patented formulation consisting of wild blueberry powder extract (89%), L-cysteine (10%), and L-glutathione (1%) were used (Table 1; hereafter referred to as WBE 111, WBE 222, WBE 444, and WBE 888). The wild blueberry extract was obtained from the *Vaccinium angustifolium* cultivar using a proprietary process. The formulation is standardized to a minimum of 35% polyphenols and contains at least 5% anthocyanins. The total polyphenols content was determined by the colorimetric Folin–Ciocalteu assay [33]. Briefly, the sample was extracted with water, and the phenolic content was determined by measuring the absorbance of the sample solution at 765 nm (after reaction with phosphotungstate-molybdate reagent in the presence of sodium carbonate to form a blue-coloured complex) and comparing it with a calibration curve using gallic acid as a standard. Anthocyanins were quantified by liquid chromatography with ultraviolet detection set at 535 nm (Agilent 1260 infinity pump and detector, Santa Clara, CA, USA), as described by Cassinese [34]. Briefly, the flow rate from the Agilent pump was 1.0 mL·min^−1^. Separation was performed on a Zorbax Extend_C18 250 mm × 4.6 mm, 5 µm column (Agilent, Santa Clara, CA, USA). Mobile phase A was water with 10% formic acid, and mobile phase B was a mixture of formic acid/methanol/acetonitrile/water (10:22.5:22.5:40, *v*/*v*). Anthocyanins were eluted with a gradient from 93–75% of A in 35 min and 75–35% of A in 10 min. The quantification was carried out by external calibration using cyanidin chloride and kuromain chloride response factor and specific molecular weight correction factor. The contents of the placebo capsules consisted of the inert components of the wild blueberry formulation, namely artificially coloured maltodextrin. All treatments were administered as three opaque, beige, size 1 capsules. All investigational products were supplied by Givaudan and manufactured according to GMP by Eurofins Amatsigroup (Saint Gely Du Fesc, France). Study products were stored in dry conditions, sheltered from light, and at room temperature.

#### 2.2.4. Screening Measures

Mini-Mental State Examination (MMSE) [35]—This is an 11-item screening tool for cognitive function where a higher score indicates better cognitive function. Scores of 24–30 indicate no impairment; scores of 18–23 indicate mild cognitive impairment; and scores of 0–17 indicate severe cognitive impairment. All participants in the current study achieved scores of 24 and above, suggesting no cognitive impairment. National Adult Reading Test (NART) [36]—The NART is a test of single-word reading consisting of 50 irregular words. Participants were asked to read aloud down the list of words, and the number of errors was recorded. All words violate grapheme–phoneme correspondence rules, meaning pronunciation cannot be determined by spelling (e.g., chord, depot). NART is used to estimate premorbid intellectual functioning, as it relies on previous word knowledge rather than current cognitive ability, making it a robust marker of intellectual attainment regardless of cognitive impairment. Memory Self-Efficacy Scale; Frequency of Forgetting (FoF) [37]—The memory self-efficacy scale, derived from the frequency of forgetting scale of the memory functioning questionnaire, is a 10-item measure used to assess memory self-efficacy. Participants were asked to rate each statement to reflect the type of memory problems they have for certain scenarios (e.g., “faces”; “where you put things”) from 1 (major problems) to 7 (no problems). Ravens Progressive Matrices (RPM) [38]—The RPM is a measure of general cognitive ability consisting of 60 sets of items where each item becomes progressively more difficult. Participants were required to select the item that fits the current set best according to pattern characteristics. A higher score reflects higher general cognitive ability. Yale Physical Activity Survey (YPAS) [39]—The YPAS is an instrument used to measure physical activity in older adult populations. The current study utilised the energy expenditure (kcals per week) outcome of this scale as a proxy for weekly physical activity.

#### 2.2.5. Cognitive Tests

Rey’s Auditory Verbal Learning task (RAVLT) [40]—Participants listened to a pre-recorded audio list of 15 words, with 1 s intervals between each word (list A), proceeded by an immediate verbal free recall. List A was presented on 5 consecutive occasions (recalls 1–5), and participants were asked to verbally recall the words they remembered after each presentation. An interference list containing 15 new words (list B) was then played to participants followed by an immediate verbal free recall of list B (recall B). Participants were then asked to recall list A after a short delay of 2 min (recall 6) and a long delay of 25 min (recall 7). Lastly, all words from lists A and B, plus 20 additional filler words were presented visually on a screen. Participants were asked to identify words from list A (word recognition) by pressing a “yes” key and new words by pressing a “no” key. Twenty-four versions of word lists were counterbalanced across practice and test sessions. All word lists consisted of 15 one-syllable (*n* = 9) and two-syllable (*n* = 6) words and were matched for familiarity (range: 525–599) and concreteness (range: 549–599). The task produces several verbal episodic memory outcome variables: total acquisition (sum recalls 1 through to 5), immediate word span (recall 1), proactive interference (recall 1 minus recall B), retroactive interference (recall 5 minus recall 6), delayed recall (recall 7), plus accuracy and speed of word recognition (number of words correctly identified), as determined by Lezak et al. [40].

Corsi Blocks task [41]—This computerised task measures spatial episodic memory. Nine stationary, square, white blocks were presented to participants on the computer screen with a black background. Participants were required to observe a sequence in which the blocks changed in colour from white to red at a rate of 1 block per second. At the end of the sequence, the instruction “Go” on the screen instructed the participants to start repeating the sequence back in the same order. The left button on the computer mouse was used for block selection. Sequence length ranged from 2 to 9 blocks, giving 8 levels of difficulty. Subjects received 4 trials per level, which were randomised across the session, with 32 trials in total. In accordance with Whyte et al. [6], measures of (1) the number of correctly reproduced sequences, (2) the longest sequence correctly remembered, and (3) first block latency (ms; time taken to commence repeating of the sequence) were recorded.

Task Switch task (TST) [28,31]—Participants viewed a circle divided into 8 equal segments, with a bold line across the horizontal diameter bisecting the circle into top and bottom halves. Numbers 1–9 (excluding 5) were randomly presented in a clockwise direction in each segment for 3000 ms with an inter-stimulus interval of 500 ms. For trials in the bottom half of the circle, participants were required to identify numbers lower than 5 using the left arrow key and numbers higher than 5 using the right arrow key on the computer keyboard. For trials in the top half, participants were required to identify odd numbers using the left arrow key and even numbers using the right arrow key. These tasks switched every 4 trials, with the switch sequence cycles denoted as S1, S2, S3, and S4. One cycle was comprised of 8 trials with 48 cycles presented to participants. Measures of accuracy (correct responses out of total trials as quotient) and response time for correct responses (ms) for task and switch sequence were acquired separately, according to Whyte et al. [28].

Trail Making tasks A and B [40]—Twenty-five circles were presented on the computer screen in a mock-random arrangement. In TMT-A, the circles contained the numbers 1–25. Participants were required to click, using the computer mouse, on the circles in ascending order. In TMT-B, 13 circles contained the numbers 1–13, and 12 circles contained the letters A–L. Participants were asked to click the cursor on circles and alternately on numbers and letters, with numbers in ascending order and letters in alphabetical order. Outcome variables included (1) time taken to complete TMT-A (ms) as a measure of visual search and motor speed and (2) time to complete TMT-B (ms) as a measurement of mental flexibility. Both TMT-A and TMT-B were not time restricted, (3) TMT B-A [40].

Choice Reaction Time task—A fixation “X” appeared on the screen for 1000 ms between presentations of the stimulus “X” on the right- or left-hand side of the screen. Interstimulus intervals of 250 ms, 500 ms, 750 ms, 1000 ms, and 1500 ms, were randomised across the session, comprising of 60 trials. Participants were required to respond as rapidly as possible by pressing the “z” key for stimuli appearing on the left or the “m” key for stimuli appearing on the right of the screen. Accuracy (correct responses out of total trials as quotient) and response time for correct responses (ms) measured facets of general alertness and motor speed, respectively.

#### 2.2.6. Mood

Positive and Negative Affect Scale (PANAS) [42]—The PANAS questionnaire was administered at the end of each cognitive test session to assess any changes in mood. The questionnaire consists of 20 mood-related items: 10 measure positive affect, and 10 measure negative affect. For each item, participants were asked to rate their present mood against a 5-point Likert scale marked from “1—very slightly or not at all” at the far left to “5—extremely” at the far-right end. A Positive Affect Score and a Negative Affect Score were generated by summing the ratings for positive and negative mood items, respectively. The possible scores ranged between 10 and 50, with a higher score corresponding to a greater level of positive or negative affect.

#### 2.2.7. Cardiovascular Measures

Blood pressure and heart rate readings were measured using an Omron M6 Comfort automatic digital blood pressure monitor (Omron Healthcare UK Ltd., Milton Keynes, UK). Measurements were taken as per the manufacturer’s instructions. All readings were taken with the cuff applied to the left upper arm between 1 and 2 cm above the elbow. Participants remained in a seated position with their feet flat on the ground and their arm supported by resting on an adjacent desk. The mean of three consecutive readings separated by 2 min intervals was recorded.

#### 2.2.8. Procedure

Data collection was conducted at the Nutritional Psychology Unit at the University of Reading School of Psychology, U.K. Participants completed all testing in individual booths. At the first screening visit, all participants gave fully informed consent and were tested for study eligibility using the MMSE, NART, FFQ, RPM, and YALE. Participants also had their height and weight measured to calculate BMI. At the second screening visit, participants completed a practice of all tasks as per the test day for familiarisation and to reduce practice effects [43]. Following allocation to treatment order, participants were booked in to attend 1 test session visit per week for the next 5 weeks. The minimum 7-day washout between sessions was strictly adhered to across the study. During the 24 h prior to each test visit, participants were required to follow a low-polyphenol diet to limit the effects of prior polyphenol consumption on cognitive and cardiovascular outcomes and to abstain from caffeinated beverages and alcohol. A 12 h dietary fast was also implemented prior to each test visit, with which all participants complied.

On arrival at 08:00 h, participants received a standardised, low-polyphenol breakfast of two croissants (total energy 338 kcal, fat 18.2 g, carbohydrate 38 g, and protein 5.2 g) with one cup of water. As in Figure 2, at 08:30 h, participants completed baseline tests of cognition and mood. Tests were administered in the following order: RAVLT (immediate recall), Choice Reaction Time, Corsi Blocks task, Trail Making tasks A and B, Task Switch task, PANAS and RAVLT (delayed recall and recognition). Blood pressure and heart rate were measured at the end of the test battery when participants had been seated for 45–60 min.

Treatment capsules were administered immediately after baseline measurements. Participants repeated the same test battery at 2, 4, and 6 h post intervention. A standardised low-polyphenol lunch consisting of a sliced white-bread cheese sandwich and a packet of ready salted crisps (total energy 400.5 kcal, fat 20.8 g, carbohydrate 48.7 g, and protein 23.7 g) was consumed immediately after the 2 h post-intervention test battery. Ad libitum consumption of water was permitted throughout the test day. On completion of the final test session, participants were debriefed and received an expenses payment of GBP 70.

### 2.3. BEAT Study 2

#### 2.3.1. Participants

A power calculation based on the pooled reaction time-effect size (f = 0.21) from the ROAB trial determined 86 participants were required to detect cognitive change at a power of 0.8 (*p* < 0.05). This sample size was unachievable due to COVID-related government restrictions implemented in March 2020, which halted further recruitment to the study. Therefore, the full sample consisted of 45 volunteers who had completed all stages of testing (see Figure 3).

#### 2.3.2. Design

The study utilised a randomized, double-blinded, placebo-controlled, cross-over design comprised of 3 visits each separated by a 7-day washout. The first visit involved eligibility assessment, screening, and familiarisation measures THAT were identical to the ROAB screening session. Following screening, participants were randomised to a treatment order by Givaudan France Naturals. On each intervention day, a standardised breakfast and lunch were provided, as in ROAB. Participants attended two test sessions across the day (see Figure 4).

Intervention: Both WBE 222 and placebo treatments were administered as a single white, opaque capsule. WBE 222 and placebo were formulated and stored as per ROAB (Table 1).

#### 2.3.3. Measures

Selection of cognitive tests for the BEAT trial was based on sensitivity to WBE in the ROAB trial. Corsi Blocks as well as TST and RAVLT were utilised (for details, see ROAB “Test Measures”). Alternate, matched versions of each task were created, and the order of trials within Corsi Blocks was counterbalanced using the Latin square method. For TST, the number of cycles presented was increased to 96 to keep the duration of test sessions similar across both studies. Blood pressure and heart rate measurements were collected as in ROAB.

#### 2.3.4. Procedure

Participants adhered to a 24 h low-polyphenol diet and a 12 h fast prior to each test day. On arrival at the lab at 08:00 h, subjects received a standardised, low-polyphenol breakfast of two croissants with water. At 08:30 h, subjects completed a test battery consisting of cognition and blood pressure measures. Testing took place in the following order: RAVLT, Corsi Blocks task, TST, blood pressure, and heart rate. The intervention was administered at 12:00 h, followed by a standardised, low-polyphenol lunch at 13:00 h, after which the final test battery was at 14:00 h (2 h after the intervention). Standardised meals were identical and contained the same amount of macronutrients as in ROAB. Upon completion of the study, participants were debriefed and received an expenses payment of GBP 50. Both studies were reviewed and given a favourable ethical opinion for conduct by the University of Reading Research Ethics Committee (ROAB UREC 18/04; BEAT UREC 19/19). The studies were performed in accordance with the current version of the declaration of Helsinki (52nd WMA General Assembly, Edinburgh, Scotland, October 2000). The trial was conducted in agreement with the International Conference on Harmonisation (ICH) guidelines on good clinical practice (GCP).

### 2.4. Analysis

Composite scores were calculated as described by Andrade [44] in both ROAB and BEAT trials to align with practice in the assessment of dementia onset or progression in older adults (Alzheimer’s Disease Composite Score; ADCOMS) [45] and in non-demented older adults (Z-scores of Attention, Verbal fluency, and Episodic memory for Nondemented older adults composite; ZAVEN) [46], of which both show sensitivity for detecting pre-clinical cognitive decline [47]. The raw data for each of the cognitive outcomes were transformed by standardising according to the standard deviation from the mean of the current study sample, giving Z scores per subject for each dependent variable. Equal weighting was given to each of the component outcomes, and the mean of Z scores were calculated to give a combined score. A global composite score was calculated based on the combination of all test measures for an indication of overall cognitive ability. Specific cognitive domains were measured by grouping outcomes according to either episodic memory or executive function (as in Table 2).

Z score analysis was used to identify outliers; datapoints with z score >3.29 were removed prior to analysis [48]. Linear mixed models (LMMs), using an unstructured covariance matrix to model successive repeat test sessions, were used to analyse data from all measures separately. Baseline performance was included as a repeated covariate to account for variation in post-treatment values that may be predicted by performance at baseline and to improve precision when estimating treatment effects [49]. Visit (ROAB: 1–5; BEAT: 1–2), dose (ROAB: WBE 111, WBE 222, WBE 444, WBE 888, or placebo; BEAT: WBE 222, or placebo), and session (ROAB only: 2 h, 4 h, and 6 h post intervention) were included as fixed factors in the model. Visit was included in order to model any residual practice effects. Post hoc, Sidak-corrected, pairwise comparisons were used to investigate any significant interactions with dose, reported to *p* < 0.05. For the Task Switch task only, the additional factors of task and switch sequence as well as their respective dose interactions and dose x session interactions were added to the model.

## 3. Results

Participant characteristics (Table 3) of both ROAB and BEAT indicated relatively healthy older adult samples within optimal BMI ranges for their age (31–32 and 27–28 kg/m2 for female and male 65 years+, respectively) [50]. No participants presented with cognitive impairment as assessed by the MMSE (all scores between 24–30), and NART mean error scores were relatively low, equivalent to an above-average IQ of 120–121. Participants scoring in the higher range of errors on NART were still deemed as having above-average IQs of 105–106, indicating the premorbid intellectual functioning of the sample was good. Mean energy expenditure (YPAS) was slightly below the mean value of a similar population in Schuler et al. [51] (combined male and female mean = 8885.5, SD = 4955.5).

### 3.1. ROAB (Study 1) Results

#### 3.1.1. Executive Function Composite Measure

Significantly poorer executive functioning was observed at 4 h compared to 2 h and 6 h (*p* < 0.05) for placebo participants only (dose x session interaction, F(8, 206) = 2.057, *p* = 0.04). This suggests a postprandial dip at 4 h for placebo treatment, which did not occur following WBE doses (Figure 5).

#### 3.1.2. Task Switch Task (TST)

A significant dose x session interaction (F(8, 1688) = 3.820, *p* < 0.001; Figure 6) revealed faster reaction time for the WBE 111 dose at 2 h (compared to WBE 888), 4 h (compared to WBE 222), and 6 h (compared to WBE 444 and WBE 222). At the 4 h timepoint, RTs were also significantly faster for WBE 888 relative to WBE 444, WBE 222, and placebo. The absence of a significant difference between WBE 111 and WBE 888 at 4 h suggests RT performance was similar across these two doses. These patterns were likely driven by dose (F(4, 1673) = 8.757, *p* < 0.001), where significantly faster reaction times were seen for WBE 111 compared to placebo and all other WBE doses. Accuracy on TST was significantly lower at 4 h compared to 2 h and 6 h (F(2, 599) = 9.819, *p* < 0.001), aligning with the postprandial dip observed in the EF composite results.

#### 3.1.3. Choice Reaction Time (CRT)

Participants elicited faster choice reaction times following WBE 888 compared to WBE 444 (F(4, 1365) = 2.745, *p* = 0.027). However, session was a significant predictor for CRT accuracy (F(2, 474) = 3.284, *p* = 0.038), where participants were more accurate at 2 h compared to 6 h, regardless of dose.

#### 3.1.4. RAVLT

When considering dose effects, WBE 444 showed superiority across retroactive interference (RI) (F(4, 211.113) = 3.001, *p* = 0.019) and word recognition RT (F(4, 304.262) = 5.793, *p* < 0.001). RI was significantly lower (better) following consumption of WBE 444 compared to WBE 111 (*p* = 0.015). No significant differences in RI between other WBE doses or placebo were observed. Word recognition RT was significantly faster with WBE 444 compared to placebo, WBE 111, and WBE 888. A reduction in performance at the 4 h session, aligning with the post-lunch dip, was also highlighted across RAVLT measures (delayed word recall (F(2, 52) = 13.944, *p* < 0.001), foil words (F(2, 53) = 5.729, *p* = 0.006), total recall (F(2, 54) = 4.097, *p* = 0.022), and RI (F(2, 53) = 3.410, *p* = 0.04). Performance was significantly worse at 4 h compared to 2 h for measures of foil words, total recall, and RI. For delayed word recall, fewer words were recalled at 4 h and 6 h compared to 2 h.

#### 3.1.5. TMT A and B, EM Composite, Corsi Blocks Task, and the Global Composite

No significant effects or interactions with dose and session were observed for TMT A, TMT B, TMT B-A, the EM composite, Corsi Blocks task, or the global composite.

#### 3.1.6. Mood—Positive (PA) and Negative Affect (NA)

Participants showed a linear decline in PA across the test day (F(2, 53) = 38.046, *p* < 0.001), with significant worsening of mood at 4 h compared to 2 h and at 6 h, suggesting changes in affect occurred alongside cognitive decline across the day. Higher NA was observed specifically at 4 h compared to 6 h and 2 h (F(2, 48) = 4.405, *p* = 0.017), implying there was a peak in negative affect at the session following lunch.

#### 3.1.7. Blood Pressure and Heart Rate

As shown in Figure 7, WBE 222 was associated with significantly lower SBP (*p* = 0.038) and DBP (*p* = 0.027) compared to placebo. DBP was also significantly lower for WBE 222 than WBE 888 (*p* = 0.025). Regardless of dose, SBP and DCP were highest at the 6 h session relative to 2 h and 4 h sessions (all *p* < 0.001). The lowest blood pressure readings were observed at the 4 h session (compared to 6 h and 2 h) for both systole and diastole (all *p* < 0.005). Heart rate was found to be significantly higher at the 4 h session compared to 2 h and 6 h, regardless of dose (F(2, 52 = 40.308, *p* < 0.001).

### 3.2. BEAT (Study 2) Results

Task Switch Task reaction time (ms) was found to be significantly faster for WBE 222 compared to placebo (F(1, 303) = 6.4, *p* = 0.01; Figure 8). The numbers of total correct sequences in the Corsi Blocks task were higher following placebo relative to WBE 222 (F(1, 29) = 4.9, *p* = 0.03). No significant treatment or visit effects were observed for composite scores of EF, EM, or global function as well as RAVLT outcomes, remaining Corsi Blocks and TST measures, or BP and heart rate (see Table 4).

## 4. Discussion

The two studies in this paper examined the acute effects of WBE on cognition, mood, and blood pressure 2–6 h post consumption in a healthy, cognitively ageing population. The collective aim was to identify an optimal WBE dose and temporal profile for cognitive benefits across the course of a day, including during the known postprandial dip where cognitive resources are low. Initially, ROAB investigated the EM, EF, and cardiovascular effects of a range of WBE doses (111 mg, 222 mg, 444 mg, and 888 mg) 2 h, 4 h, and 6 h following a single one-off administration. Benefits were seen for episodic memory following the WBE 444 dose, with faster recognition and protection against retroactive interference, whilst faster executive function was observed following the WBE 111 dose in the Task Switch task. Collectively, these results suggest WBE may have the potential to improve the speed of processing across some EM and EF measures at low doses. Additionally, results from ROAB support the expected postprandial 14:00 h dip [22,24], exemplified by reductions in EF at 4 h post intervention for placebo only. Interestingly, the absence of an EF dip following WBE doses at this timepoint suggests that WBE may help to maintain cognitive resources during this critical window.

This aligns with previous research across child [16], young adult [28], and middle-aged adult [19] populations, where berry treatments alleviated the natural decline in cognition across the course of a day. Further support for a postprandial cognitive dip was evidence by a reduction in episodic memory at the 4 h timepoint. However, this dip was observed regardless of dose, suggesting WBE may not have been sensitive to auditory verbal learning domains in the same way as EF. Strikingly, WBE 222 produced acute reductions in both systolic and diastolic blood pressure compared with placebo, regardless of specific timepoint. BEAT followed on from ROAB by investigating the effects of a single treatment of WBE 222 on EM, EF, and blood pressure during the anticipated post-lunch dip at 14:00. Notwithstanding faster EF for WBE 222 compared to placebo, the largely null findings of BEAT indicate limited cognitive or blood pressure benefits of WBE 222 in healthy older adults under these conditions. The absence of time-related reductions in performance at 2 h suggests participants may not have experienced the postprandial circadian dip in cognition shown in ROAB, which may explain the largely null findings in BEAT. The absence of the cognitive dip could be due to an increased rest period and decreased cognitive burden (fewer test sessions) during BEAT, resulting in participants who were less cognitively fatigued relative to ROAB. Potentially, the conditions were not suitably challenging for WBE 222 to show sensitivity to postprandial and circadian effects. Aligned with this, we saw a relatively high withdrawal rate in the ROAB study, with 12 participants withdrawing after attending the familiarization visit where participants experienced a full day of testing and realized the demands of both the task battery and length of the test day. Although circumstantial, this does provide further support to indicate that ROAB induced a much greater degree of cognitive fatigue compared to BEAT, where withdrawal rates were much lower.

Future research should aim to explore the acute metabolic profile of WBE specifically, especially in formulas that contain additional compounds such as L-cysteine and L-glutathione, which enhance the stability of anthocyanins and increase bioavailability. Any future investigations should endeavour to provide a control condition containing the stabilizing ingredients alongside the placebo to ensure that any effects can be attributed directly to the WBE. Such an investigation would allow researchers to more closely match cognitive test schedules to when WBE-specific metabolites are most likely to be circulating in the bloodstream and then manipulate cognitive demand elements (for example, lunch consumption, rest, fatigue, and test schedule and characteristics) to ascertain optimal WBE conditions.

It is interesting to observe that faster reaction times were seen across both ROAB and BEAT studies for the Task Switch task at 111 mg and 222 mg, respectively. Reassuringly, this was not at a cost to accuracy, meaning participants were able to accurately complete trials at a faster speed under these WBE doses. In support, previous research that employed this task in cognitively ageing adults under a chronic regimen of daily freeze-dried blueberry (24 g/day) equivalent to 1 cup of fresh blueberries [31] also found beneficial effects. The TST has been suggested to be particularly sensitive to acute WBE intervention due to its high demand and inhibitory nature, a task characteristic for which flavonoids have shown to be previously sensitive [16,17,19,28,29]. Collectively, this highlights the need for targeted research to assess flavonoid sensitivity in specific cognitive domains and populations during conditions of cognitive fatigue.

Consistent effects on the TST in both acute and chronic studies support the assumption that flavonoids may have similar biological mechanisms of action across acute and chronic timeframes. One of the purported mechanisms is through increased cerebral blood flow (CBF), mediated by improved endothelial function. Flavonoids may increase nitric oxide (NO) production and in turn promote NO-mediated vasodilation [52,53]. Such effects are mediated through endothelial function (see review [52]), as demonstrated by improvements to flow-mediated dilation (FMD) [13,54,55]. Akazawa et al. [56] demonstrated a significant correlation between endothelial function and cerebral blood flow (CBF) in a cohort of middle-aged men, implying the two may be mechanistically linked. It is therefore plausible that benefits to endothelial function may subsequently improve CBF and thus cognitive function by increasing the delivery of metabolic substrates required for brain function. Such vasoactive actions may be associated with reductions in blood pressure relating to flavonoid consumption (see review [57]). Blood pressure, FMD, and CBF may therefore be mechanistically linked to cognitive function by way of endothelial-dependent vasodilation. It is therefore interesting that the ROAB trial observed subtle cognitive improvement alongside blood pressure reductions, indicating potential support for this mechanism.

ROAB demonstrated clear reductions in both systolic and diastolic blood pressure with WBE 222 compared to placebo; however, this result was not replicated using WBE 222 in BEAT. Post hoc *t*-tests comparing the baseline blood pressure of the two sets of subjects showed significantly higher (*p* = 0.037) baseline diastolic blood pressure for BEAT (mean = 75.75 mmHg; SD = 9.15 mmHg) compared to ROAB (mean = 70.97 mmHg; SD = 9.55 mmHg). Systolic BP readings followed a similar pattern, but differences between BEAT (mean = 122.85 mmHg; SD = 15.47 mmHg) and ROAB (mean = 116.66 mmHg; SD = 15.95 mmHg) baselines were non-significant. These results suggest that blood pressure responses to WBE 222 may be dependent on initial blood pressure readings. Specifically, WBE 222 may be more sensitive in reducing BP for individuals with lower initial BP readings. Considering the CBF–endothelial function hypothesis discussed previously, it would, theoretically, make sense that acute flavonoid action may be most efficient in those with better baseline (more responsive) endothelial function. It would be interesting to observe BP change following acute WBE 222 in a population with poor endothelial function to test this hypothesis further.

## 5. Conclusions

Acute WBE supplementation was observed to improve the reaction times of healthy older adults in an executive function task during the post-lunch dip. These effects were consistent across ROAB and BEAT trials. Similarly, improved reaction times in relation to episodic memory and significant alleviation of the post-lunch decline in cognitive performance compared to placebo were observed in ROAB but not BEAT. These findings suggest post-lunch testing of older adults may optimise the circadian decline in cognitive performance, especially in executive function measures, which may create cognitively demanding conditions optimal for WBE sensitivity. The level of cognitive fatigue imposed by the study design and its impact on outcomes should be accounted for in future research as well as exploration into the metabolite profiles of berry extract formulas that contain stabilising agents. A significant lowering of both systolic and diastolic blood pressures relative to placebo was also observed in ROAB but not BEAT. The initial blood pressure readings of an individual in combination with the demands of the test day schedule may influence the cardiovascular response to WBE supplementation. However, there were a large number of outcome measures across the two trials, and the effects were subtle, with benefits only observed on a small number of outcomes. Nevertheless, the advantages that were observed could be particularly meaningful for older-aged adults for whom small improvements in reaction time and BP may enhance daily cognitive functioning and health. Further research is required to explore the utility of WBE in populations already experiencing mild cognitive impairments.

## Figures and Tables

**Figure 1 nutrients-16-01180-f001:**
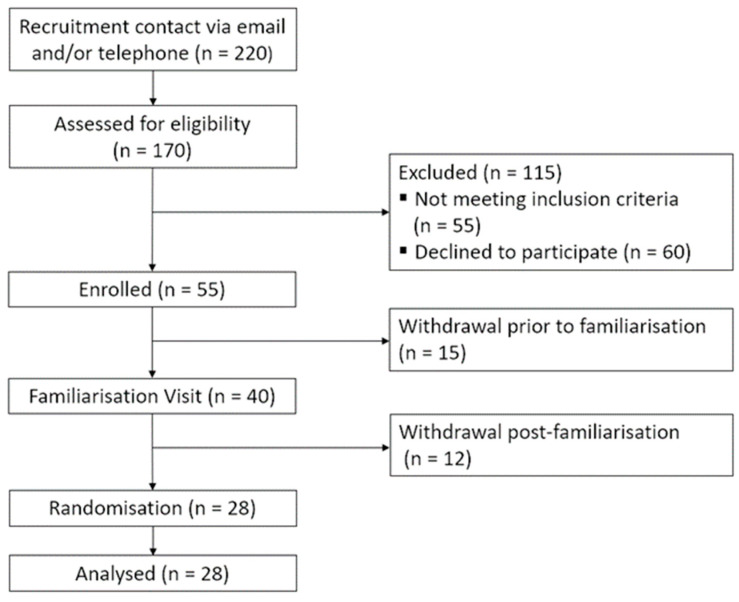
Consort diagram of the participant sample at each stage of recruitment and participation in the ROAB study.

**Figure 2 nutrients-16-01180-f002:**
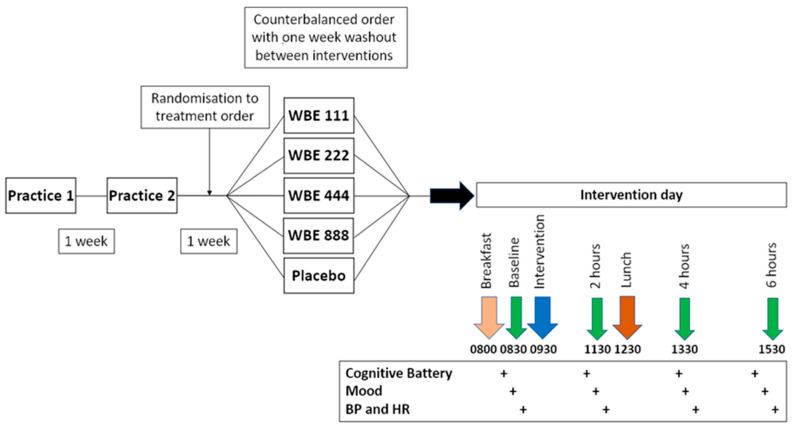
ROAB study design and timeline of procedure on test days.

**Figure 3 nutrients-16-01180-f003:**
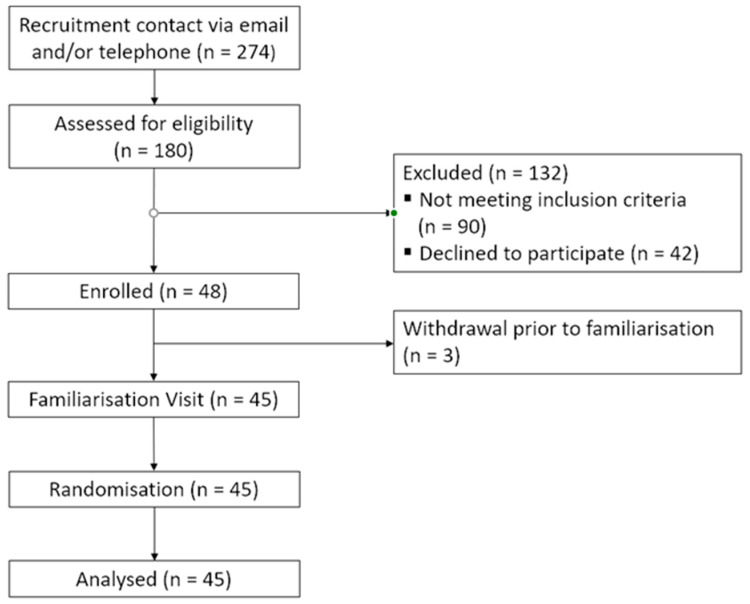
Consort diagram of participant sample at each stage of recruitment and participation for the BEAT study.

**Figure 4 nutrients-16-01180-f004:**
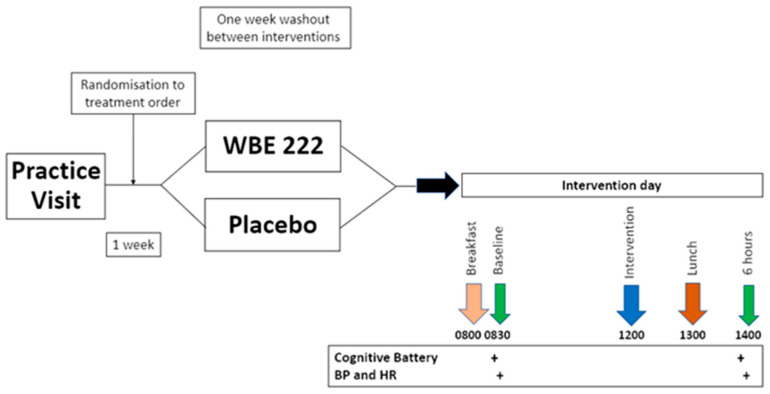
BEAT study design and timeline of procedure on test days.

**Figure 5 nutrients-16-01180-f005:**
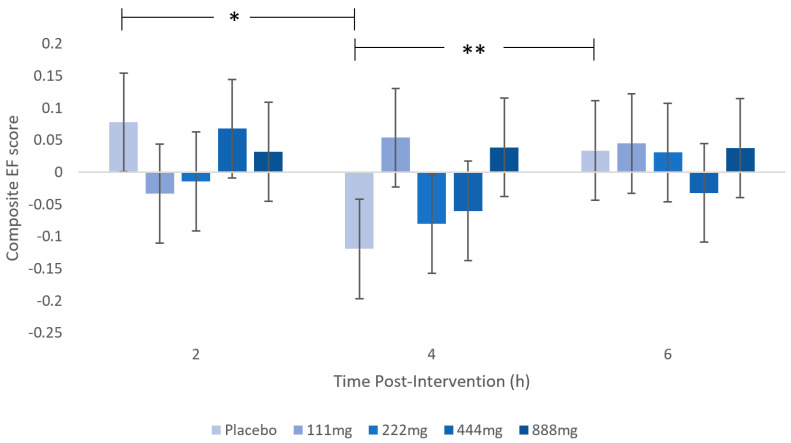
Executive Function Composite scores at each session (2 h, 4 h, and 6 h) following each dose (111, 222, 444, 888, or placebo). A significant dip in performance is seen at 4 h with placebo intervention but not with WBE supplementation. * *p* < 0.05; ** *p* < 0.01.

**Figure 6 nutrients-16-01180-f006:**
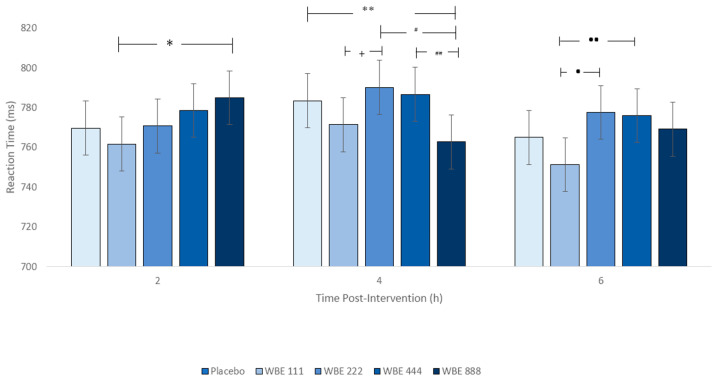
Mean (± SE) Task Switch task reaction time by dose and session. * *p* = 0.002; ** *p* = 0.008; # *p* < 0.01; ## *p* = 0.001; + *p* = 0.024; • *p* < 0.001; •• *p* = 0.001.

**Figure 7 nutrients-16-01180-f007:**
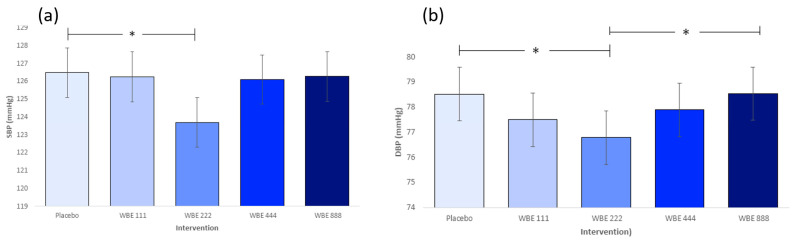
(**a**) Mean systolic blood pressure as a function of intervention (± SE), showing significantly lower SBP with WBE 222 than placebo. (* *p* = 0.038). SBP (F(4, 231.331) = 3.069, *p* = 0.017). (**b**) Mean diastolic blood pressure as a function of intervention (± SE), showing significantly lower DBP with WBE 222 than placebo (* *p* = 0.027) and WBE 222 than WBE 888 (* *p* = 0.025). DBP (F(4, 210.271) = 3.223, *p* = 0.014).

**Figure 8 nutrients-16-01180-f008:**
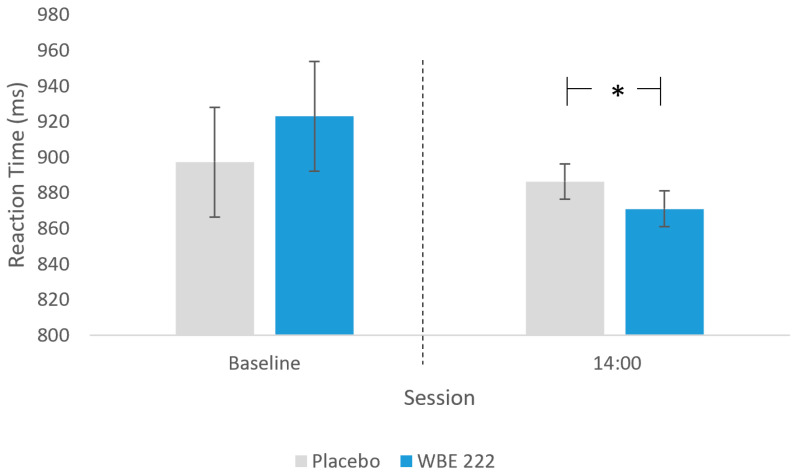
Mean Task Switch Task RT (± SE) as a function of treatment, showing significantly faster reaction times with WBE 222 than placebo (* *p* = 0.012). Baseline performance, included as a covariate in the analysis, is shown separated by the dotted line for reference.

**Table 1 nutrients-16-01180-t001:** Ingredients of each wild blueberry extract (WBE) treatment intervention and placebo.

Condition	Intervention Formulation
Placebo	Inert artificially coloured maltodextrin (300 mg), once daily, 3-hard-capsule regimen
WBE 111	100 mg WBE + 11 mg L-cysteine + 1 mg L-glutathione, once daily, 3-hard-capsule regimen
WBE 222	200 mg WBE + 22 mg L-cysteine + 2 mg L-glutathione, once daily, 3-hard-capsule regimen
WBE 444	400 mg WBE + 44 mg L-cysteine + 4 mg L-glutathione, once daily, 3-hard-capsule regimen
WBE 888	800 mg WBE + 88 mg L-cysteine + 8 mg L-glutathione, once daily, 3-hard-capsule regimen

**Table 2 nutrients-16-01180-t002:** Cognitive components of each composite score for ROAB and BEAT trials.

Composite Score	ROAB Cognitive Measures	BEAT Cognitive Measures
Global	All test measures	All test measures
Episodic memory	Corsi longest sequence remembered, Corsi total number of correct sequences remembered, Corsi latency, RAVLT word recognition accuracy, RAVLT delayed word recall accuracy, foil words rejected, RAVLT words learned, RAVLT total recall, and RAVLT PI, RAVLT RI, and RAVLT word recognition reaction time	Corsi longest sequence remembered, Corsi total number of correct sequences remembered, Corsi latency, RAVLT word recognition accuracy, RAVLT delayed word recall accuracy, RAVLT total recall, and RAVLT PI, RAVLT RI, and RAVLT word recognition reaction time
Executive function	CRT accuracy, Switching task accuracy, and TMT B-A, TMT A, TMT- B, CRT, and TST reaction time	Switching task accuracy and Switching task reaction time

**Table 3 nutrients-16-01180-t003:** Demographic data across both ROAB and BEAT trials. M = male, F = female.

Baseline Characteristics	ROAB (*n* = 28)		BEAT (*n* = 45)	
	Mean	SD	Mean	SD
Age at enrolment (years)	70.71	2.14	71.02	2.03
Gender (M:F)	11:17	-	18:27	-
BMI (kg/m^2^)	24.51	3.30	25.05	2.95
MMSE (/30)	27.9	1.58	28.53	1.38
NART (number of errors/50)	9	5.11	8.04	4.44
Frequency of Forgetting 1	5.01	0.75	5.27	0.98
Frequency of Forgetting 2	5.22	1.10	5.36	1.19
Raven’s Progressive Matrices	50.21	5.86	48.98	7.03
Yale expenditure (kcal/week)	5519	3798.36	6050	2384.61
Energy consumption (kcal/day)	1772	632.85	1904	778.82
Fruit consumption (g/day)	291	172.61	339	241.25
Vegetable consumption (g/day)	299	179.15	339	136.41

**Table 4 nutrients-16-01180-t004:** Mean (SE) of BEAT outcomes following WBE 222 and placebo. Statistical differences from placebo are indicated * *p* < 0.05.

Measure	Intervention	Timepoint		*p*-Value 2 h Comparison
		Baseline (Covariate)	2 h	
Executive Function Composite	Placebo	0.30 (0.09)	−0.010 (0.04)	
	WBB 222	−0.46 (0.09)	0.003 (0.04)	0.52
Task Switch Task Accuracy	Placebo	0.97 (0.006)	0.97 (0.003)	
	WBB 222	0.97 (0.006)	0.97 (0.003)	0.16
Task Switch Task RT (ms)	Placebo	897 (31)	886 (10)	
	WBB 222	923 (31)	871 (10)	0.01 *
Episodic Memory Composite	Placebo	−0.31 (0.08)	0.011 (0.03)	
	WBB 222	0.18 (0.08)	−0.01 (0.03)	0.53
Word Recognition Accuracy	Placebo	0.85 (0.02)	0.80 (0.02)	
	WBB 222	0.88 (0.02)	0.80 (0.02)	0.99
Word Recognition RT (ms)	Placebo	1215 (51)	1235 (39)	
	WBB 222	1201 (51)	1206 (39)	0.52
Delayed Word Recall	Placebo	8.79 (0.55)	6.51 (0.40)	
	WBB 222	9.22 (0.54)	6.55 (0.40)	0.94
Total Recall	Placebo	49.82 (1.44)	46.31 (0.99)	
	WBB 222	50.63 (1.45)	45.42 (0.99)	0.53
Proactive Interference	Placebo	0.94 (0.05)	1.03 (0.05)	
	WBB 222	0.95 (0.05)	1.01 (0.05)	0.74
Retroactive Interference	Placebo	2.38 (0.30)	3.02 (0.26)	
	WBB 222	2.36 (0.32)	2.91 (0.34)	0.7
Immediate Recall	Placebo	6.84 (0.25)	5.80 (0.21)	
	WBB 222	6.65 (0.25)	5.96 (0.21)	0.49
Corsi Longest Sequence	Placebo	7.35 (0.22)	7.39 (0.19)	
	WBB 222	6.98 (0.22)	7.35 (0.19)	0.85
Corsi Total Correct Sequences	Placebo	0.52 (0.02)	0.54 (0.02)	
	WBB 222	0.53 (0.02)	0.51 (0.02)	0.03 *
Corsi Latency	Placebo	1017 (30)	1013 (21)	
	WBB 222	1014 (30)	1023 (21)	0.72
Systolic BP (mm/Hg)	Placebo	123.00 (2.40)	124.54 (1.34)	
	WBB 222	123.47 (2.40)	124.18 (1.35)	0.81
Diastolic BP (mm/Hg)	Placebo	79.41 (4.43)	75.49 (0.95)	
	WBB 222	77.30 (4.42)	75.47 (0.95)	0.98

## Data Availability

The data that support the findings will be available in the University of Reading Research Data Archive at https://researchdata.reading.ac.uk/ following an embargo from the date of publication to allow for commercialization of research findings.

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
