# Peer review of "Wild Blueberry Extract Intervention in Healthy Older Adults: A Multi-Study, Randomised, Controlled Investigation of Acute Cognitive and Cardiovascular Effects"

_nutrients, 2024, doi:10.3390/nu16081180_

Round 1

Reviewer 1 Report

Comments and Suggestions for Authors

The paper is well written, there are very few studies which have been conducted to date looking at this expansive range of dose response of berries and exploring these endpoints over such a wide range of time points postprandially. The study design and methodology is appropriate and conclusions are not overstated.

Few minor suggestions.

In instances where the authors describe the berry extract doses its not always clear what they are referring too. For example - does this mean 111mg of extract containing 7mg of anthocyanins when they state (111, 7mg)? If so this should be clearly spelled out at least in the first instance of use of this nomenclature.

-The methods table indicates a significant amount of cysteine which suggests this is as much a cysteine trial as a anthocyanin or blueberry trial and it would have been appropriate to suggest in the discussion that a suitable control condition (containing placebo + cysteine) would be required to investigate doses without this potentially significant confounding variable.

_ A possible link between vasodilation and cognition could have been described in the introduction. It comes out in the discussion but in the introduction its unclear why vascular measures are included and what potential interaction effects there may be.

-In the study flow diagram there is a high withdrawl rate in the first study- is this normal? Any comment as to why this may be, in the discussion section could be useful to the reader?

-There is no methodology describing for how the analytes in composition table one were established? What methods were used? Are these off-label or measured by the company or in-house or taken from a database? Characterization should be described for any intervention products in studies.

Author Response

We thank the reviewer for their positive views on our manuscript.

  1. In instances where the authors describe the berry extract doses its not always clear what they are referring too. For example - does this mean 111mg of extract containing 7mg of anthocyanins when they state (111, 7mg)? If so this should be clearly spelled out at least in the first instance of use of this nomenclature.

We thank the reviewer for their suggestion and confirm that in the first instance of the use of the nomenclature we have amended the text as suggested to include the ‘dose’ and the ‘anthocyanin content’ [Page 2, Lines 55-57], we have also added further clarification of our nomenclature in the methods [Page 5, Line182-183]

  1. The methods table indicates a significant amount of cysteine which suggests this is as much a cysteine trial as a anthocyanin or blueberry trial and it would have been appropriate to suggest in the discussion that a suitable control condition (containing placebo + cysteine) would be required to investigate doses without this potentially significant confounding variable.

We agree that we should have mentioned this point and have added text in the discussion as requested: Any future investigations should endeavor to provide a control condition containing the stabilizing ingredients, alongside the placebo to ensure that any effects can be attributed directly to the WBE. [Page 16, Lines 565-567]

  1. A possible link between vasodilation and cognition could have been described in the introduction. It comes out in the discussion but in the introduction its unclear why vascular measures are included and what potential interaction effects there may be.

We have added a brief mention of the link between cognition and vascular function, and some exemplar references, to help introduce the vascular measures we have used in the study: Improvements in vascular function have been hypothesized to underlie the improve-ment in cognitive performance, for example through changes in cerebral blood flow (CBF) [1-4]. [Page2, Lines 48-50]

  1. In the study flow diagram there is a high withdrawal rate in the first study- is this normal? Any comment as to why this may be, in the discussion section could be useful to the reader?

The rate of withdrawal in Study 1 is quite high, and was certainly higher than we have seen with other berry interventions trials that we have run in the laboratory. We have added some discussion of this point to the Discussion: Aligned with this, we saw a relatively high withdrawal rate in the ROAB study with 12 participants withdrawing after attending the familiarization visit; where participants experienced a full day of testing and realized the demands of both the task battery and length of the test day. Although circumstantial this does provide further support to in-dicate that ROAB induced a much greater degree of cognitive fatigue compared to BEAT where withdrawal rates were much lower. [Page 16, Lines 557-562]

  1. There is no methodology describing for how the analytes in composition table one were established? What methods were used? Are these off-label or measured by the company or in-house or taken from a database? Characterization should be described for any intervention products in studies.

We thank the reviewer for their suggestion and confirm that we have added some relevant details in the methods section of the manuscript: The formulation is standardized to a minimum of 35% polyphenols and contains at least 5% anthocyanins. The total polyphenols content is determined by the colorimetric Fo-lin-Ciocalteu assay [33]. Briefly, sample is extracted with water and the phenolic content is determined by measuring the absorbance of the sample solution at 765 nm (after re-action with phosphotungstate-molybdate reagent in the presence of sodium carbonate to form a blue-colored complex) and comparing it with a calibration curve using gallic acid as a standard. Anthocyanins are quantified by liquid chromatography with ultra-violet detection set at 535 nm (Agilent 1260 infinity pump and detector) as described by Cassinese [34]. Briefly, the flow rate from the Agilent pump is 1.0 ml.min-1. Separation is done on a Zorbax Extend_C18 250 x 4.6mm, 5µm column. Mobile phase A is water with 10% formic acid and mobile phase B is a mixture of formic ac-id:methanol:acetonitrile:water (10,22.5,22.5,40, v/v). Anthocyanins are eluted with a gradient from 93-75% of A in 35 minutes and 75-35% of A in 10 minutes. The quantifi-cation is done by external calibration using cyanidin chloride and kuromain chloride response factor and specific molecular weight correction factor. The contents of the placebo capsules consisted of the inert components of the wild blueberry formulation, namely artificially coloured maltodextrin. All treatments were administered as three beige opaque size 1 capsules. All investigational products were supplied by Givaudan and manufactured according to GMP by Eurofins Amatsigroup (Saint Gely Du Fesc, France). Study products were stored in dry conditions, sheltered from light, at room temperature.  [Page 5, Lines 184-204]

Reviewer 2 Report

Comments and Suggestions for Authors

The paper describes the effect of acute WBE supplementation in healthy older adults. The ROAB trial demonstrated a significant alleviation of the post-lunch decline in cognitive performance and a significant lowering of both systolic and diastolic blood pressure.

The Authors demonstrated the benefical effect of  wild blueberry extract on cognitive function and blood pressure in older adults.
The manuscript is well written, it reports the results of two trials conducted on this topic, the references are appropriate and the conclusions are consistent with the evidence and arguments presented.  Tables and figures are clearly presented.

Author Response

We thank the reviewer for their very positive views on our manuscript.

Reviewer 3 Report

Comments and Suggestions for Authors

The work presents the results of very methodologically extensive research. Unfortunately, little is revealed from this study.

The design of the study did not ensure specificity - the observed slight improvement in cognitive functions at the selected dose may be due to the properties of the extract, the presence of additional antioxidants, improvement in the function and condition of the endothelium via the effect on blood pressure - too many factors that may affect the expected effect, as well as the effect is very negligible, to enrich your knowledge about the properties of WBE.

No data on the qualitative and quantitative profile of the extracts, in terms of the content of active substances, as well as the lack of comparative studies on WBE with and without the addition of antioxidants, decrease the role of the experimental design itself. There is also no description of the possible dependence of the effects on the duration of WBE supplementation (from the 1st to the 5th week).

A better idea would be to start the project with research involving patients with mild cognitive impairment, which the authors proposed as a continuation of those presented in this work. The results could be more spectacular and encouraging.

Detailed reservations:

1)      Lack of justification for the range of WBE doses used in the case of ROAB.

2)      It seems that selection of WBE 222 mg for BEAT tests is controversial – this dose in EF, TST, RAVLT or CRT tests was not the most effective at all –more comprehensive justification is needed.

3)      The unclear way of presenting the results (a graphical presentation supported by a description would be more clear).

Author Response

We thank the reviewer for their comments and advice on repeating the study with patients with mild cognitive decline. We agree that the results from such a study may prove more encouraging, however we believe that our manuscript describes a methodologically sound set of studies and, whilst the data show few cognitive effects from the WBE, we do not believe that we have overstated our findings and as such still believe that such data deserve publishing. The comments we received from reviewers 1 and 2 also support this view. 

  1. Lack of justification for the range of WBE doses used in the case of ROAB.

We thank the reviewer for their suggestion and agree that we need to do more to justify the doses chosen. We have added some relevant text in the introduction: The lowest dose used here (WBE 111 containing 7mg anthocyanins) was the dose previously established to produce beneficial cognitive effects and reduction in SBP following 3 months daily treatment [10]. As suggested by a recent meta-analysis on dietary flavonoids and cognition [30], it was predicted that cognitive effects would increase dose-dependently and be most pronounced for the highest dose of WBE (WBE 888 con-taining 56mg anthocyanins). [Page 3, Lines 124-130]

  1. It seems that selection of WBE 222 mg for BEAT tests is controversial – this dose in EF, TST, RAVLT or CRT tests was not the most effective at all –more comprehensive justification is needed.

Given that one of the leading mechanisms to explain the cognitive benefits of flavonoid-rich interventions is via the vascular system, we chose to utilise the dose that had shown the most significant vascular effects from the ROAB study when designing the BEAT study. We have attempted to clarify this position by a more explicit description of the interaction between cognition and vascular function [Page2, Lines 48-50; as requested also by Reviewer 1] and also by adding explicit mention of the vascular effects in ROAB being the dominant reason for the choice of dose in BEAT: (WBE 222mg, chosen for the significant effects on blood pressure shown in ROAB) [Page 3, Lines 139-140]. We hope these additions provide sufficient explanation of our decisions.

  1. The unclear way of presenting the results (a graphical presentation supported by a description would be more clear).

We are unsure what the reviewer requires here, we believe we have presented the data in the clearest manner possible. Our rationale was as follows, for the ROAB study, we have provided figures for the cognitive outcomes only when a significant dose x session effect was evident, any main effects of dose or treatment are described clearly in the text without additional figures. We do not think that the manuscript would be improved by providing additional figures of these main effects. For the blood pressure outcome, given the rationale for choosing this dose in BEAT, a graph of the main effect of dose was included. In the BEAT study, we chose to include figures of the main significant effects; a table was included to outline all of the other data not in the figures. We believe the text fully describes all of the findings. We are happy to add further figures/graphs or description if the reviewer requests it.